# Metabolomic Profiling of Adipose Tissue in Type 2 Diabetes: Associations with Obesity and Insulin Resistance

**DOI:** 10.3390/metabo14080411

**Published:** 2024-07-26

**Authors:** Argyri Mathioudaki, Giovanni Fanni, Jan W. Eriksson, Maria J. Pereira

**Affiliations:** Department of Medical Sciences, Clinical Diabetes and Metabolism, Uppsala University, 75185 Uppsala, Sweden; argyri.mathioudaki@medsci.uu.se (A.M.); giovanni.fanni@medsci.uu.se (G.F.); jan.eriksson@medsci.uu.se (J.W.E.)

**Keywords:** metabolomics, adipose tissue, type 2 diabetes, insulin sensitivity, glucose uptake

## Abstract

The global prevalence of Type 2 Diabetes (T2D) poses significant public health challenges due to its associated severe complications. Insulin resistance is central to T2D pathophysiology, particularly affecting adipose tissue function. This cross-sectional observational study investigates metabolic alterations in subcutaneous adipose tissue (SAT) associated with T2D to identify potential therapeutic targets. We conducted a comprehensive metabolomic analysis of SAT from 40 participants (20 T2D, 20 ND-T2D), matched for sex, age, and BMI (Body Mass Index). Metabolite quantification was performed using GC/MS and LC/MS/MS platforms. Correlation analyses were conducted to explore associations between metabolites and clinical parameters. We identified 378 metabolites, including significant elevations in TCA cycle (tricarboxylic acid cycle) intermediates, branched-chain amino acids (BCAAs), and carbohydrates, and a significant reduction in the nucleotide-related metabolites in T2D subjects compared to those without T2D. Obesity exacerbated these alterations, particularly in amino acid metabolism. Adipocyte size negatively correlated with BCAAs, while adipocyte glucose uptake positively correlated with unsaturated fatty acids and glycerophospholipids. Our findings reveal distinct metabolic dysregulation in adipose tissue in T2D, particularly in energy metabolism, suggesting potential therapeutic targets for improving insulin sensitivity and metabolic health. Future studies should validate these findings in larger cohorts and explore underlying mechanisms to develop targeted interventions.

## 1. Introduction

The global prevalence of Type 2 Diabetes (T2D) has reached pandemic proportions, posing a significant public health challenge due to its association with severe complications such as cardiovascular diseases, neuropathy, and nephropathy [1]. A pathophysiological hallmark of T2D is insulin resistance, a metabolic derangement characterized by the diminished ability of cells to respond to insulin, primarily affecting adipose tissue, liver, and muscle [2]. The adipose tissue plays a critical role in regulating whole-body glucose homeostasis by storing excess energy as triglycerides and releasing free fatty acids during fasting periods. However, in T2D, the adipose tissue becomes dysfunctional, leading to abnormal glucose and fat turnover [3,4].

Understanding the metabolic alterations within the adipose tissue in T2D is crucial for developing novel therapeutic strategies. Metabolomics, a powerful technique for analyzing small molecule profiles, offers a comprehensive view of cellular metabolism. By investigating the biochemical alterations associated with T2D in adipose tissue, it is possible to identify potential biomarkers for early diagnosis, elucidate mechanisms underlying insulin resistance [5], and develop targeted drugs to tackle its development.

Insulin resistance is characterized by a specific metabolic signature featuring higher levels of circulating branched-chain amino acids (BCAA) and disrupted acylcarnitine metabolism during fasting [6,7]. Abnormal serum metabolome response to an oral glucose load is also evident in individuals with T2D [8], supporting the concept of metabolic inflexibility in the insulin-resistant state. However, much less is known about the adipose tissue metabolomic signature in T2D. BCAA and other amino acids were shown to be associated with hyperglycemia in the visceral adipose tissue (VAT) metabolome of individuals with T2D [9]. The subcutaneous adipose tissue (SAT) content of non-esterified fatty acid is associated with in vivo glucose uptake of the VAT, as assessed by ^18^F-FDG-PET [10]. Interestingly, the SAT metabolome does not differ between individuals with obesity and with and without T2D, who are candidates for obesity surgery [11]. In a targeted metabolomic study, ex vivo consumption of pyruvate and pyroglutamate was higher in individuals with prediabetes and obesity compared to those without obesity, while acetate production was lower in individuals with prediabetes and obesity compared to subjects without obesity [12]. To the best of our knowledge, no study has ever deployed an untargeted metabolomic approach on the SAT to highlight differences between individuals with and without T2D and obesity.

This exploratory study aimed to provide a detailed metabolomic analysis of SAT from T2D patients compared to subjects without T2D, focusing on identifying metabolites associated with adipocyte glucose uptake and cell size. By elucidating the metabolic profile of adipose tissue in T2D, we hope to contribute to a better understanding of insulin resistance and identify potential therapeutic targets.

## 2. Materials and Methods

### 2.1. Study Participants

This study utilized a cross-sectional observational design to investigate the metabolic alterations in SAT in subjects with and without T2D. Forty participants were recruited between 2013 and 2014 from the outpatient clinical research unit at Uppsala University Hospital. The participants were divided into two groups: 20 with T2D and 20 without T2D (ND), matched for sex, age, and body mass index (BMI). Each group was further subdivided based on obesity status (with and without obesity), as detailed in Appendix A. T2D diagnosis was confirmed according to the American Diabetes Association (ADA) criteria. All participants with T2D had been receiving metformin treatment for at least 3 months prior to the study, with dosages ranging from 500 mg to 2500 mg as per clinical guidelines. Detailed participant characteristics and glucose uptake data in isolated adipocytes have been previously reported [4] and are summarized in Table 1.

The study protocol was approved by the Regional Ethics Review Board in Uppsala (Dnr 2013/183 and 2013/494). All participants provided written informed consent.

### 2.2. Study Procedures

After an overnight fast, a medical assessment was initially performed, and detailed anthropometric measurements were obtained. Then, fasting blood samples were collected and analyzed for plasma glucose levels, lipid profile, serum insulin levels, and C-peptide concentrations by the hospital’s Department of Clinical Chemistry. Next, an adipose tissue needle biopsy was performed on the lower abdominal SAT after local anesthesia with lidocaine (Xylocaine; AstraZeneca, Södertälje, Sweden). Portions of the adipose tissue were rapidly frozen in liquid nitrogen for subsequent metabolomic analyses, and part was transported to the laboratory for isolation of adipocytes and ex vivo measurement of glucose uptake. To assess glucose tolerance, participants underwent an oral glucose tolerance test (OGTT) by ingesting a 75 g glucose solution. Blood samples were drawn throughout the OGTT to measure plasma glucose and fatty acids (FFA) and serum insulin levels. The area under the curve (AUC) was calculated using the trapezoidal rule. 

### 2.3. Glucose Uptake in Isolated Adipocytes, Ex Vivo

Adipocytes were isolated and glucose uptake was measured, as previously reported [4]. In brief, adipose tissue samples were digested in a collagenase solution to isolate adipocytes. These adipocytes were then washed in a glucose-free Krebs-Ringer bicarbonate medium. For glucose uptake measurements, adipocytes were incubated in this medium at 37 °C, both without insulin (basal) and with 25 or 1000 µU/mL of insulin. After incubation with D-(U-14C) glucose, the cells were separated from the media by centrifugation and the radioactivity was measured to assess basal and insulin-stimulated glucose uptake. The capacity of glucose uptake in adipocytes was assessed by calculating two distinct ratios: the glucose uptake at 1000 µU/mL insulin (representing the maximal response) divided by the basal (no insulin), and the glucose uptake at 25 µU/mL insulin (representing insulin sensitivity) divided by basal glucose uptake.

### 2.4. Adipose Tissue Metabolomics

Metabolite quantification in adipose tissue was carried out using Metabolon Inc.’s TrueVision™ (Durham, NC, USA) analysis, which encompasses the global mVision platform, as previously described [4,8]. The analyses utilize GC/MS and LC/MS/MS platforms for comprehensive profiling. Metabolite levels were expressed in arbitrary units. Metabolites that were missing in more than 20% of the samples within both the subjects without T2D and T2D groups were excluded from the analyses (modified 80% rule) [13] (Appendix A). For the remaining metabolites that did not reach the detection threshold, missing values were imputed using the minimum observed value for that metabolite. 

### 2.5. Statistical and Enrichment Analyses

No formal power analyses were performed as this was an exploratory study. Clinical characteristics were compared between subjects with and without T2D using the Mann-Whitney U test. Principal Component Analysis (PCA) was used to explore the clustering patterns of metabolites within the adipose tissue samples. PCA was conducted using the PCA function in Python from the *sci-kit-learn* library, specifying two principal components to capture the maximum variance in a two-dimensional space. The distributions of metabolite concentrations between the subjects with and without T2D were compared with Mann-Whitney using the *dplyr* [14], and *stats* [15] packages in R version 4.2.3. Metabolites with *p*-values less than 0.05 were further examined in enrichment and pathway analyses. Spearman correlations with adipocyte size and adipocyte insulin-stimulated glucose uptake were conducted in R with a custom function. Correlated metabolites were visualized with bar plots in GraphPad Prism version 10.2.2 for Windows (GraphPad Software, Boston, MA, USA), to examine the direction and strength of correlations and the category type of the correlated metabolites. Enrichment and pathway analysis were performed with R package *MetaboAnalystR* 4.0 [16]. These tests were based on the global test [17] and the metabolite library used was the Relational database of Metabolomic Pathways (RaMP_DB) that includes 3694 metabolites from multiple sources such as KEGG and Reactome to provide detailed annotations and enrichment analyses for proteins, and metabolites queries [18].

## 3. Results

### 3.1. Overview of the Analysed Metabolite Panel

The metabolomic analyses from both subjects with and without T2D revealed the presence of 378 analytes in SAT, including various metabolic classes such as lipids, amino acids, peptides, carbohydrates, nucleotides, cofactors and vitamins, energy metabolites, and xenobiotics (Figure 1a).

Principal Component Analysis (PCA) revealed notable overlap among the metabolite groups (Figure 1b). However, PCA analyses for specific metabolic classes showed similar overlap patterns across most metabolite groups, except for the energy metabolites, which exhibited clear differentiation based on T2D status (see Appendix A). 

### 3.2. Metabolic Alterations in Adipose Tissue in Subjects with and without T2D and Obesity

We examined differences in the concentration distributions of these 378 SAT metabolites among the study groups, pooled by metabolic class (Figure 2). Subjects with both T2D and obesity showed the most significant alterations (Figure 2, red), with elevated levels of amino acids, peptides, and carbohydrates, compared to all other groups. Additionally, subjects with T2D and obesity had significantly higher energy-related and lipid levels compared to subjects without T2D. Subjects with T2D but without obesity also showed higher amino acid levels and peptides compared to subjects without T2D, regardless of obesity status.

### 3.3. Comparison of Adipose Tissue Metabolites between Subjects with and without T2D

Thirty-eight metabolites were significantly different in individuals with T2D compared to those without T2D (Figure 3 and Appendix A). Individuals with T2D exhibited significantly higher levels of metabolites involved in the TCA cycle, such as alpha-ketoglutarate, gluconate, and malate. Several amino acids also showed significant differences between the groups, with higher levels of isoleucine, leucine, lysine, methionine, *N*-acetylaspartate, serine, hydroxyproline, and 2-aminoadipate in individuals with T2D. Glucose levels were also significantly increased in the adipose tissue of individuals with T2D, consistent with the known metabolic characteristics of the disease. Other notable changes included elevated levels of mannose, glycerate, and phosphoenolpyruvate, and decreased levels of the nucleotide-related metabolites including adenine, adenosine, adenosine monophosphate (AMP), and cytidine monophosphate (CMP) in SAT in individuals with T2D compared to those without T2D. Lipids showed minimal changes, with elevated levels of oleamide and 1-arachidonoyl-GPI (20:4) and reduced levels of myristoleic acid in individuals with T2D compared to those without T2D. Metformin levels were notably higher in individuals with T2D, reflecting its therapeutic use and accumulation in adipose tissue.

We also wanted to elucidate whether distinct patterns would emerge when subdividing the subjects by obesity status (Figure 3 and Appendix A). 

(i) Amino Acids and Peptides: T2D individuals, particularly those with obesity, exhibited higher levels of specific amino acids and peptides compared to subjects without T2D. (ii) Carbohydrates and Energy Metabolites: Subjects with T2D, regardless of obesity status, showed elevated levels of glucose, glycerate, mannose, phosphoenolpyruvate (PEP), and key TCA cycle intermediates like alpha-ketoglutarate and malate, indicating significant disruptions in carbohydrate and energy metabolism. (iii) Lipids: Significant changes in lipid profiles were observed, with T2D individuals, especially those with obesity, showing elevated levels of specific lipids such as 1-arachidonoyl-GPI (20:4) and oleamide. (iv) Cofactors and Vitamins: Higher levels of gamma-tocopherol and alpha-tocopherol were found in T2D subjects, particularly in those with obesity. (v) Nucleotides: Adenosine levels were lower in T2D without obesity compared to subjects without T2D and without obesity, indicating potential disruptions in nucleotide metabolism. (vi) Xenobiotics: Elevated levels of gluconate were found in T2D individuals with obesity, reflecting increased oxidative stress.

### 3.4. Correlation Analyses between Metabolites in Adipose Tissue and Clinical Characteristics

Correlation analyses between the identified metabolites and clinical markers of adiposity, hyperglycemia, and insulin resistance were performed (Appendix A) and the significant associations are presented in Figure 4. No specific patterns of correlations could be found between metabolite categories and clinical variables. Lipids such as cholesterol and 2-myristoylglycerol (14:0) were positively correlated with BMI. In contrast, specific fatty acids such as FA18:3n3 (a-Linolenic acid) and FA20:5n3 (Eicosapentaenoic acid) were negatively correlated with BMI and WHR (Waist-to-Hip Ratio). Additionally, FA18:3n3 negatively correlated to plasma insulin and OGTT AUC insulin and positively with the insulin sensitivity index Matsuda. The specific lipid metabolite oleate (18:1n9) showed a positive correlation with HbA1c and OGTT AUC for FFA. Amino acids including isoleucine and leucine showed a positive correlation with WHR. Some amino acids, like leucine, serine, and tryptophan negatively correlated with OGTT AUC insulin. The peptides like tryptophylglycine and tyrosylglutamate were negatively correlated with BMI. Looking into the carbohydrates category, glucose and mannose were positively correlated with multiple markers of hyperglycemia and insulin resistance, such as fasting plasma glucose, HbA1C, serum insulin, OGTT AUC glucose, HOMA-IR (Homeostatic Model Assessment for Insulin Resistance), and WHR. Alpha-tocopherol and energy metabolites like alpha-ketoglutarate and gluconate showed positive correlations with markers of hyperglycemia (fasting plasma glucose and HbA1c and OGTT AUC glucose). Alpha-tocopherol also correlated with markers of insulin resistance including HOMA-IR and serum insulin. 

### 3.5. Associations between Adipose Tissue Metabolites and Adipocyte Size

The correlation analysis between adipose tissue metabolites and adipocyte size revealed 118 significant associations across different metabolite classes (Figure 5a and Appendix A). Lipids and amino acids exhibited the highest number of correlations, with 19 and 54 metabolites, respectively. Most lipids, amino acids, and peptides demonstrated negative correlations with adipocyte size, indicating lower levels in subjects with larger adipocytes. A few positive correlations were observed (*n* = 6) within the lipids, amino acids, and xenobiotics categories. Specifically, amino acids such as serine, threonine, and valine, along with lipids like dehydroisoandrosterone sulfate (DHEA-S) and glycerol 3-phosphate (G3P), showed negative correlations with adipocyte size. Conversely, lipids such as FA14:1n5 (myristoleic acid) and FA18:1n7 (vaccenic acid) exhibited positive correlations. Additionally, carbohydrates like glucose, phosphoenolpyruvate (PEP), glycerate, and 3-phosphoglycerate, as well as energy metabolites including malate, acetylphosphate, alpha-ketoglutarate, and succinate, displayed negative correlations with adipocyte size. Over-representation analysis revealed these correlated metabolites are involved in pathways related to valine, leucine, and isoleucine biosynthesis, alanine, aspartate, and glutamate metabolism, and arginine biosynthesis (Figure 5b).

### 3.6. Associations between Adipose Tissue Metabolites and Adipocyte Glucose Uptake

Correlation analyses with glucose uptake after incubation with 1000 µU/mL of insulin, reflecting the maximal insulin response relative to basal levels, revealed 25 significantly correlated metabolites (Figure 6a and Appendix A). Higher levels of certain amino acids, including 3-methylglutaconate, 3-indoxyl sulfate, isobutyrylcarnitine (C4), and gamma-glutamylglutamine, were negatively associated with maximal glucose uptake, while only creatine showed a positive correlation. Among the lipids category, several metabolites, such as FA18:3n3 (a-Linolenic acid), FA18:2n6 (Linoleic acid), and various glycerophospholipids including 1-linoleoyl-GPC (18:2) and 2-linoleoyl-GPE (18:2), exhibited positive correlations with maximal glucose uptake, whereas 2-arachidonoyl-GPC (20:4) had a negative correlation. Peptides like gamma-glutamyl glutamine and tyrosylvaline also showed negative correlations with glucose uptake. No significant correlations were observed for carbohydrates, cofactors, nucleotides, and xenobiotics. Pathway enrichment analyses highlighted key metabolic pathways associated with maximal glucose uptake, including the biosynthesis of unsaturated fatty acids, linoleic acid metabolism, alpha-linolenic acid metabolism, ether lipid metabolism, and glycerophospholipid metabolism (Figure 6b). Notably, 20 of these correlations were consistent with glucose uptake after incubation with 25 µU/mL of insulin (Appendix A).

## 4. Discussion

In this study, we present the metabolic signature of T2D in the SAT, and its associations with clinical markers of insulin resistance, adipocyte size, and glucose uptake. Notably, subjects with T2D—especially those with obesity—displayed elevated levels of amino acids, including branched-chain amino acids (BCAAs) like leucine and isoleucine, peptides, and carbohydrates. These findings corroborate our prior research documenting similar T2D metabolomic signatures in plasma [8]. We also observed significant associations between this metabolic signature and markers of adiposity and glucose homeostasis, alongside correlations with SAT adipocyte size and glucose uptake. Elevated lipids and BCAAs were linked to higher adiposity, while anti-inflammatory fatty acids and certain peptides showed negative correlations. Interestingly, larger adipocytes exhibited lower amino acid levels, suggesting an increased catabolic rate. Crucially, pathways such as unsaturated fatty acid biosynthesis were positively associated with adipocyte insulin-stimulated glucose uptake. Our study provides novel metabolic profiling of SAT, which has been less studied compared to plasma or VAT, and contributes to new knowledge in the field, particularly regarding the metabolic alterations in SAT associated with T2D.

### 4.1. Metabolomic Distinctions in Adipose Tissue between Subjects with and without T2D

The PCA plot revealed an overlap between T2D and ND groups, indicating fewer metabolic differences in adipose tissue compared to plasma-based studies [19,20]. Notably, energy metabolites in SAT demonstrate a clearer distinction based on T2D status, emphasizing disruptions in energy metabolism in adipose tissue as a key characteristic of T2D. This assumption aligns with previous studies noting altered levels of energy metabolites, including increased glycolytic intermediates, in T2D [21,22]. 

Subjects with both T2D and obesity exhibited significant metabolic alterations in adipose tissue, highlighted by elevated levels of amino acids (including BCAAs like leucine, and isoleucine), peptides, carbohydrates (e.g., glucose, mannose), and energy-related metabolites (e.g., alpha-ketoglutarate, malate, gluconate). These findings are in line with earlier studies in plasma that have identified increased levels of BCAAs, aromatic amino acids, and disrupted lipid metabolism, alongside elevated glycolytic intermediates and TCA cycle metabolites [23,24,25]. The association of elevated BCAAs with diabetes and insulin resistance in plasma and skeletal muscle has been well-documented for decades [6,7,9,21,26], typically linked to impaired catabolism. Previous studies in VAT and SAT have also shown elevated levels of amino acids including BCAAs, tryptophan, and serine in subjects with hyperglycemia and T2D [9,10,27].

Interestingly, some metabolic signatures found in SAT in this study are novel, such as elevated levels of TCA intermediates (alpha-ketoglutarate, malate, and gluconate) and reduced nucleotide metabolites (adenine, adenosine, AMP, and CMP), which have not been reported in previous adipose tissue metabolomics studies in T2D subjects. Changes in these metabolites can impact energy metabolism [28,29], and contribute to metabolic dysregulation observed in T2D. 

The presence of elevated metformin levels in T2D individuals reflects its therapeutic use and accumulation in adipose tissue. Notably, all subjects with T2D in this study were taking metformin, which has been shown to impact the levels of various metabolites, including those involved in the TCA cycle, glucose metabolism, and lipid metabolism [30]. Therefore, it is important to consider that the observed metabolic differences may be influenced by both the presence of T2D and the effects of metformin therapy, making it challenging to separate the impact of the disease from the effects of the treatment. 

### 4.2. Associations with Clinical Parameters

The positive association of BCAAs like isoleucine and leucine with central adiposity is consistent with studies implicating these amino acids in obesity and metabolic syndrome through their roles in protein synthesis and energy metabolism [24,31]. Elevated levels of specific lipids, such as certain fatty acids, have been shown to induce adipocyte hypertrophy and insulin resistance [32]. Elevated specific lipids with hyperglycemia, such as oleate, have been associated with lipotoxicity, contributing to insulin resistance by impairing insulin signaling pathways [33]. Conversely, negative correlations of fatty acids like a-linolenic acid and eicosapentaenoic acid with BMI and WHR align with their known anti-inflammatory and lipid-lowering effects, which may protect against fat accumulation [34,35]. 

The strong positive correlations of glucose, mannose, alpha-ketoglutarate, and gluconate with markers of hyperglycemia and insulin resistance underscore the central role of carbohydrates in metabolic dysregulation in diabetes. Elevated glucose and mannose levels are direct indicators of impaired glucose metabolism, a hallmark of T2D [19]. Alpha and gamma-tocopherol, the major isomers of Vitamin E, were positively correlated with several markers of hyperglycemia and insulin resistance. The reasons for higher levels of these metabolites with higher insulin resistance are unknown but might indicate higher dietary intake, as the major site for alpha-tocopherol storage is in adipose tissue [36]. 

### 4.3. Adipocyte Size and Glucose Uptake in Relation to Metabolomics

Our findings reveal a significant negative correlation between adipocyte size and several amino acids, particularly those involved in the biosynthesis of valine, leucine, and isoleucine, as well as alanine, aspartate, and glutamate metabolism. This result appears counterintuitive since amino acids are typically elevated in T2D and obesity, conditions often characterized by larger adipocytes. One possible explanation is that the catabolism of BCAAs drives adipogenesis, potentially leading to the recruitment of new adipocytes for differentiation [37]. Interestingly, a study demonstrated that mice with adipose tissue knockout of the enzyme that catabolizes BCAAs to brain-chain keto acids, are resistant to high-fat-induced obesity, and have reduced adipose tissue and smaller adipocytes, indicating a critical role of BCAA catabolism in regulating adipocyte size and function [38]. Future studies should investigate the enzymes involved in BCAA metabolism in human adipose tissue and whether changes in those impact adipocyte metabolism. Additionally, it is important to note that our T2D group tended to have smaller adipocytes compared to the ND group; therefore, these findings should be validated in larger cohorts to confirm their generalizability. The negative correlations of key energy metabolites like alpha-ketoglutarate and malate with adipocyte size suggest that smaller adipocytes might rely more on efficient energy production pathways. Larger adipocytes might be less efficient in utilizing and synthesizing amino acids, contributing to systemic metabolic imbalances. Mechanisms underlying the negative correlation between adipocyte size and amino acid levels may involve alterations in metabolic pathways regulated by insulin and nutrient signaling, such as mTOR signaling [39].

In turn, several lipids positively correlated with glucose uptake, such as FA18:3n3 (a-Linolenic acid), FA18:2n6 (Linoleic acid), and various glycerophosphocholines (GPCs) and glycerophosphoethanolamines (GPEs). These lipids may enhance membrane fluidity and facilitate insulin signaling, consistent with studies showing that polyunsaturated fatty acids and glycerophospholipids maintain insulin sensitivity and metabolic health [40]. These results also seem to be consistent with a previous study that identified that glycerophosphatidylcholines were a predictor to identify “insulin-sensitive obesity” in VAT [41]. Adipocyte glucose uptake was also positively correlated with creatine. Evidence has suggested that creatine supplementation may improve glucose metabolism [42] and one of the proposed mechanisms is suggested to be by increasing GLUT4 expression [34]. Negative correlations of certain amino acids, such as 3-methylglutaconate, 3-indoxyl sulfate, and isobutyrylcarnitine, with maximal glucose uptake may reflect alterations in aminoacid-mediated signaling pathways that negatively affect glucose metabolism. For example, 3-indoxyl sulfate has been associated with mitochondrial dysfunction, oxidative stress, and impaired energy metabolism [43]. Similarly, elevated tyrosylvaline might contribute to insulin resistance by affecting insulin signaling pathways [5].

### 4.4. Future Studies

Future research should focus on longitudinal studies to establish causal relationships between specific metabolic changes in adipose tissue and the progression of T2D. Investigating the mechanistic pathways underlying the observed associations between metabolites and clinical parameters will be crucial. Additionally, expanding the metabolomic analysis to include other tissues and fluids, such as muscle and blood, could provide a more comprehensive understanding of systemic metabolic changes in T2D. Integrating multi-omics approaches, including genomics, proteomics, and metabolomics, will offer deeper insights into the complex metabolic networks involved in T2D and obesity. Lastly, exploring the therapeutic potential of targeting key metabolic pathways identified in this study, such as unsaturated fatty acid biosynthesis, glycerophospholipid metabolism, and nucleotide-related metabolites could lead to novel treatments for improving insulin sensitivity and metabolic health in T2D.

### 4.5. Limitations

This study has several limitations. First, the cross-sectional and descriptive design limits the ability to infer causality between metabolic alterations and T2D. Second, the sample size, though adequate for an explorative approach, may not capture the full heterogeneity of metabolic responses in a diverse population. Third, the reliance on adipose tissue biopsies from a specific anatomical location (SAT) may not reflect the metabolic state of other fat depots such as VAT. Additionally, the study did not account for potential confounding factors such as diet, physical activity, and medication use other than metformin. 

## 5. Conclusions

In conclusion, our study reveals distinct metabolic alterations in adipose tissue associated with T2D, particularly in energy metabolism. Elevated levels of TCA cycle intermediates such as alpha-ketoglutarate, fumarate, and malate, along with BCAAs and carbohydrates, and reduced nucleotide-related metabolites indicate extensive metabolic dysregulation in SAT of T2D subjects. Obesity further exacerbates these changes, especially in amino acid metabolism. Adipocyte size is negatively associated with several BCAAs, while adipocyte glucose uptake is primarily positively associated with unsaturated fatty acid biosynthesis and glycerophospholipid metabolism. Identifying these key metabolic pathways suggests potential therapeutic targets for improving insulin sensitivity and overall metabolic health in T2D and obesity. Future research should validate these findings in larger and more diverse cohorts, elucidate underlying mechanisms, and explore the therapeutic potential of modulating specific metabolic pathways.

## Figures and Tables

**Figure 1 metabolites-14-00411-f001:**
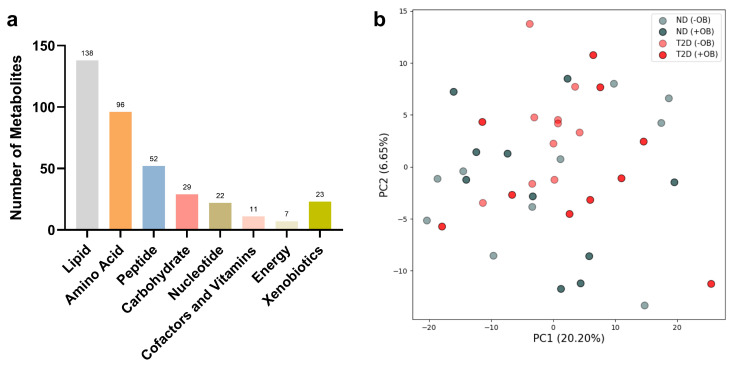
Principal Component Analysis (PCA) in Adipose Tissue annotated with T2D and Obesity Status. (**a**) Number of metabolites used for PCA analyses in each metabolic class in SAT from subjects with and without T2D. (**b**) PCA plot reveals the clustering of adipose tissue samples from subjects with or without T2D (ND) and with or without obesity (+/− OB). Each point represents an individual sample, with color coding based on T2D status (red for subjects with T2D, grey without T2D) and different shades indicating obesity status (darker for individuals with obesity, lighter for individuals without obesity). The axes represent the first two principal components (PC1 and PC2), accounting for the highest percentage of variance.

**Figure 2 metabolites-14-00411-f002:**
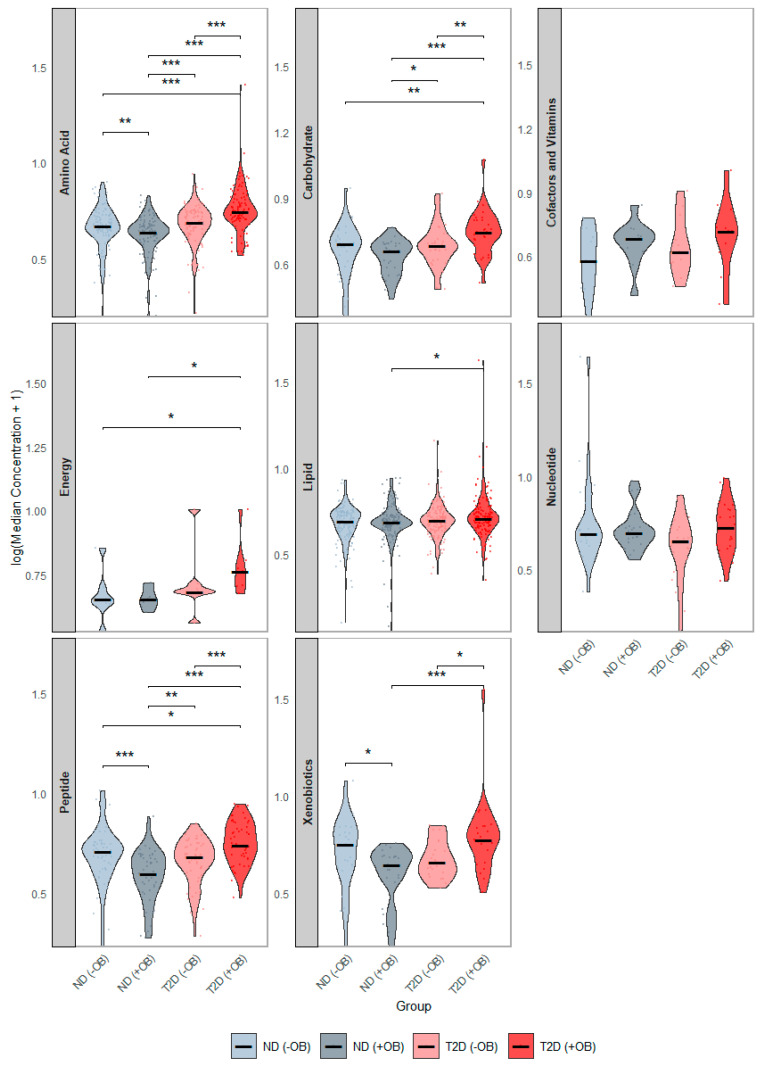
Distribution of Median Metabolite Concentrations across groups defined by Diabetes and Obesity Status. This figure presents violin plots showing the distribution of median metabolite concentrations across the groups defined by T2D and Obesity status. Each panel represents the distribution of a separate metabolic category. The four groups are annotated with distinct colors: subjects without Type 2 Diabetes (T2D) and without obesity (ND −OB), subjects without T2D but with obesity (ND +OB), subjects with T2D but without obesity (T2D −OB), and subjects with both T2D and obesity (T2D +OB). Significant Wilcoxon rank-sum tests are indicated as follows: * *p* < 0.05; ** *p* < 0.01; *** *p* < 0.001.

**Figure 3 metabolites-14-00411-f003:**
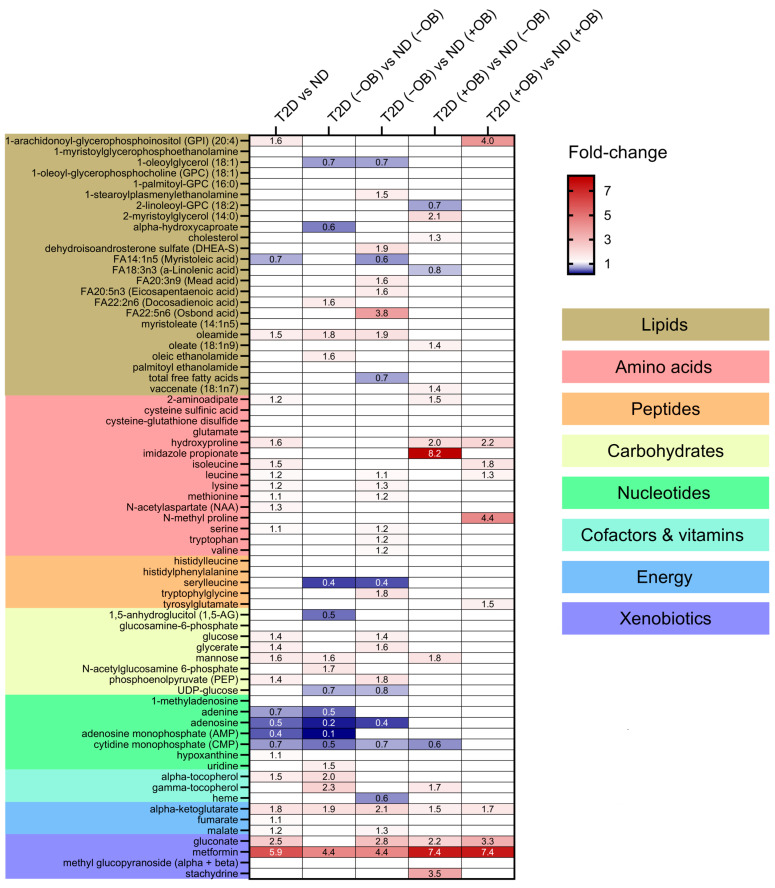
Differential metabolite levels in Adipose Tissue in Subjects with and without T2D and Obesity. Heatmap representing the relative abundance (fold change) of the 38 significantly different metabolites identified in a comparison of subjects with and without T2D and ND individuals and sub-group analyses between four groups defined by both T2D and obesity status: subjects without T2D and obesity, ND (−OB); individuals without T2D and with obesity, ND (+OB); individuals with T2D and without obesity, T2D (−OB); and individuals with both T2D and obesity, T2D (+OB). Metabolites are grouped and colored according to their category type: lipids, amino acids, peptides, carbohydrates, nucleotides, cofactors and vitamins, energy-related metabolites, and xenobiotics, with color-coded sections indicating these classes. The color scale indicates the fold change in metabolite levels, with red indicating an increase and blue indicating a decrease between groups.

**Figure 4 metabolites-14-00411-f004:**
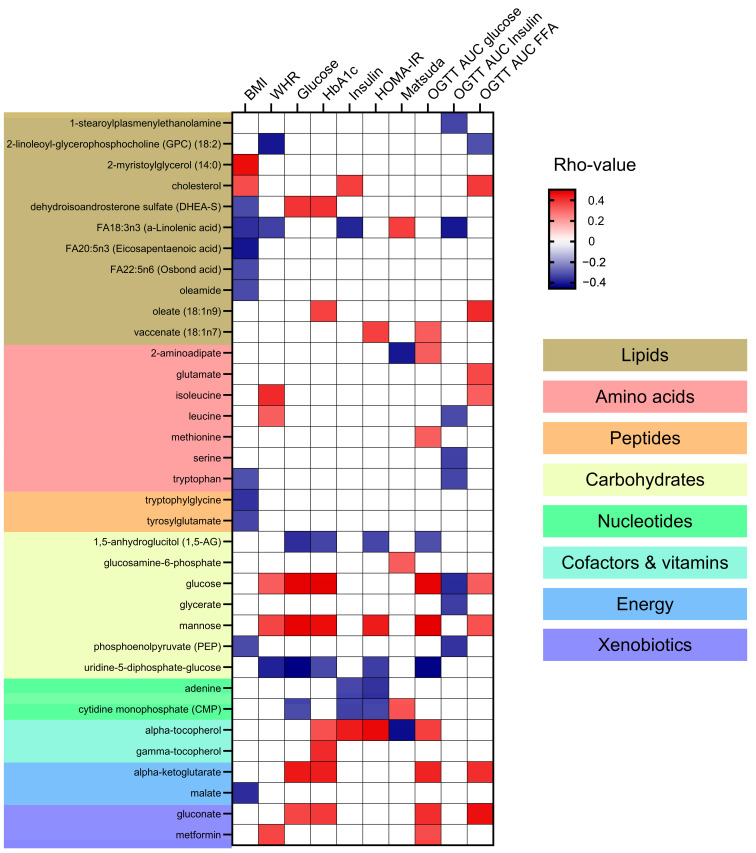
Correlation of Metabolites with Clinical Markers of Adiposity and Insulin Resistance. This heatmap presents the Spearman correlation coefficients (rho-values) between metabolites and clinical markers of adiposity and insulin resistance in this cohort of subjects with and without Type 2 Diabetes (T2D). Clinical markers include BMI (Body Mass Index), WHR (Waist-to-Hip Ratio), blood glucose levels, HbA1c (glycated hemoglobin), blood insulin levels, HOMA-IR (Homeostatic Model Assessment for Insulin Resistance), Matsuda Index for insulin sensitivity, and OGTT AUC (Oral Glucose Tolerance Test Area Under the Curve) for glucose, insulin, and free fatty acids. Positive correlations are indicated in red, negative correlations in blue, and non-significant correlations are indicated in white. Metabolites are categorized into lipids, amino acids, peptides, carbohydrates, nucleotides, cofactors and vitamins, energy, and xenobiotics, with corresponding background colors.

**Figure 5 metabolites-14-00411-f005:**
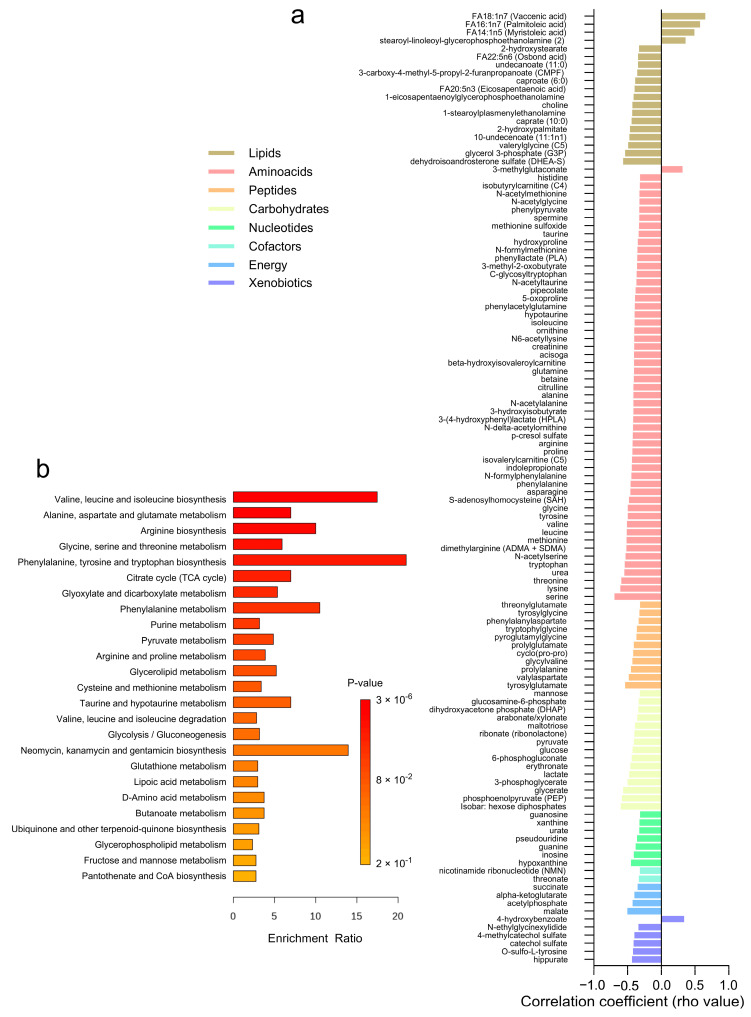
Correlation Analysis of Adipose Tissue Metabolites with Adipocyte Size. (**a**) A heatmap displaying the correlation coefficients (rho-values) of individual metabolites with adipocyte size. (**b**) A bar graph showing the enrichment ratio of the top metabolic pathways significantly associated with adipocyte size.

**Figure 6 metabolites-14-00411-f006:**
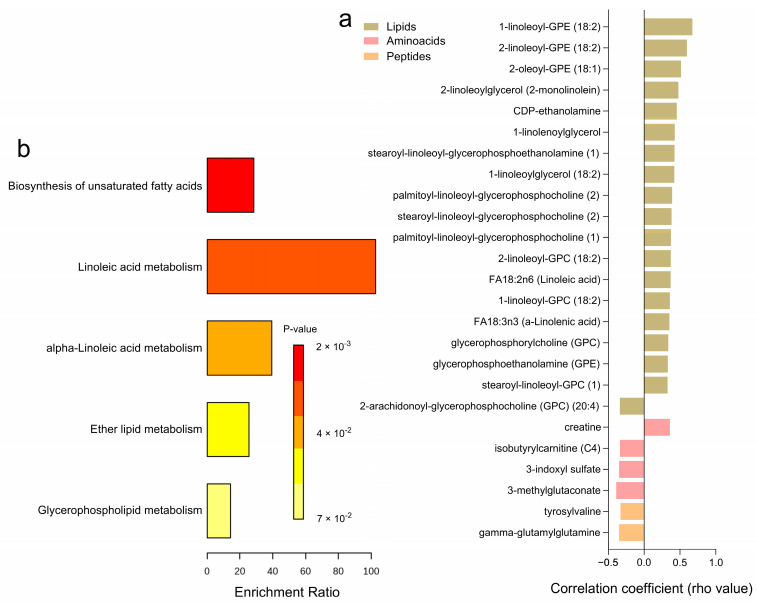
Correlation Analysis of Adipose Tissue Metabolites with Adipocyte Glucose Uptake (1000 µU/mL). (**a**) A heatmap displaying the correlation coefficients (rho-values) of individual metabolites significantly correlated with maximum adipocyte glucose uptake capacity (1000 µU/mL of insulin. relative to basal) (**b**) A bar graph showing the enrichment ratio of the top metabolic pathways significantly associated with adipocyte glucose uptake.

**Table 1 metabolites-14-00411-t001:** Clinical characteristics of study participants.

	Without T2D (*n* = 20)	T2D (*n* = 20)
N (women/men)	10/10	10/10
Age (years)	58 ± 11	58 ± 9
BMI (kg/m^2^)	30.8 ± 4.6	30.7 ± 4.9
WHR	0.96 ± 0.07	0.99 ± 0.05
Fasting plasma glucose (mmol/L)	6.0 ± 0.7	8.2 ± 1.5 ***
HbA1c (mmol/mol)	37.3 ± 3.7	48.8 ± 8.6 ***
Serum insulin (mIU/L)	11.5 ± 5.2	15.5 ± 7.0 *
HOMA-IR	3.08 ± 1.58	5.26 ± 2.86 **
Matsuda index	4.04 ± 2.11	2.65 ± 1.38 *
Adipocyte glucose uptake, basal (fL/cell/s)	37.1 ± 20.7	24.1 ± 9.3 *
Adipocyte glucose uptake, 1000 µU/mL insulin (fL/cell/s)	72.8 ± 44.8	41.2 ± 21.1 *
Maximal Glucose Uptake (fold change) ^a^	1.94 ± 0.55	1.72 ± 0.51
Adipocyte size (µm)	109 ± 10	106 ± 11
AUC OGTT glucose (mmol/L × min)	1416 ± 340	2493 ± 522 ***
AUC OGTT insulin (mIU/L × min)	10,654 ± 6276	8072 ± 3981
AUC OGTT FFA (µmol/L × min)	23,947 ± 5644	31,236 ± 8558 **

Data are presented as mean ± SD. * *p* < 0.05; ** *p* < 0.01; *** *p* < 0.001 relative to subjects without T2D (Mann-Whitney U test). BMI, Body Mass Index; WHR, Waist-to-Hip Ratio; HbA1c, glycated hemoglobin; HOMA-IR, homeostasis index of insulin resistance. ^a^ Maximal glucose uptake capacity was calculated by dividing the glucose uptake at 1000 µU/mL insulin by the basal glucose uptake.

## Data Availability

The datasets generated during and/or analyzed during the current study are not publicly available due to a collaboration agreement with a company but are available from the corresponding author upon reasonable request.

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
