# Peer review of "Metabolomic Profiling of Adipose Tissue in Type 2 Diabetes: Associations with Obesity and Insulin Resistance"

_metabolites, 2024, doi:10.3390/metabo14080411_

Round 1

Reviewer 1 Report

Comments and Suggestions for Authors

The article titled "Metabolomic Profiling of Adipose Tissue in Type 2 Diabetes: Associations with Adipocyte Insulin Sensitivity" identified with the reference code metabolites-3101611, and submitted for review in Metabolites (ISSN 2218-1989), aimed to provide a detailed metabolomic analysis of adipose tissue from T2D patients compared to subjects without T2D with a focus on identifying metabolites associated with adipocyte insulin sensitivity.

The introduction provides a comprehensive background and includes all relevant references. It effectively sets the stage for the research by offering necessary context and acknowledging previous studies, which strengthens the foundation of your work.

While the research design has potential, there are some areas that need improvement to ensure its appropriateness. Specifically, the study lacks a detailed description of its 

prospective observational design. Additionally, since this is an observational study, it should follow the STROBE guidelines for reporting to enhance its clarity and reliability. 

Addressing these points will strengthen the research design and make the study more robust and credible.

The results are clearly presented. The data is organized in a logical and easy-to-follow manner, with appropriate use of tables and figures to enhance understanding.

The conclusions are well-supported by the results. The findings are effectively summarized and directly linked to the data presented, demonstrating a clear and logical progression from the results to the conclusions. This strong alignment reinforces the credibility of the study's outcomes. 

Additional comment:

Table 1 woman/man ratio is not presented corectly. WHR is  not explained under this table.

Author Response

1.     Specifically, the study lacks a detailed description of its prospective observational design. Additionally, since this is an observational study, it should follow the STROBE guidelines for reporting to enhance its clarity and reliability. Addressing these points will strengthen the research design and make the study more robust and credible.

Response: Thank you for pointing out the need for a detailed description of the study design. We have indicated the cross-sectional observational design in the "Materials and Methods" section (page 2, lines 65-71) and abstract to enhance clarity (page 1, lines 11). We appreciate your suggestion to adhere to the STROBE guidelines. We have revised the manuscript to align with the STROBE guidelines for observational studies.

2.     Table 1 woman/man ratio is not presented correctly. WHR is not explained under this table.

Response: Thank you for bringing this to our attention. We have corrected the presentation of the woman/man ratio in Table 1 and provided an explanation for WHR in the table legend (pages 2-3).

Reviewer 2 Report

Comments and Suggestions for Authors

The study explores metabolic changes in subcutaneous adipose tissue (SAT) related to Type 2 Diabetes (T2D), focusing on insulin resistance and its implications for public health. Analyzing SAT samples from 40 participants (20 with T2D, 20 without but at risk), researchers used GC/MS and LC/MS/MS to identify 378 metabolites. They found elevated TCA cycle intermediates, branched-chain amino acids (BCAAs), and carbohydrates in T2D subjects, exacerbated by obesity. Correlations showed BCAAs linked to adipocyte size, and glucose uptake tied to unsaturated fatty acids and glycerophospholipids. These insights suggest potential targets for improving insulin sensitivity and managing T2D, urging further research validation and mechanistic exploration.

The article is well structured, with an appropriate number of experiments that are carefully designed and accompanied by sound statistical analysis. Nevertheless, there are several other details that I have listed:

1.       Line 30: Due to the significance of the assertion and to ensure relevance to the current context, it would be advisable to update the citation, which is over 7 years old.

2.       Table 1: Review the units in Table 1 for Fasting Plasma Glucose; they should be expressed as mmol/L.

3.       Line 137: It would be clearer to specify that these analytes are from both individuals with T2D and ND, as the comparison between these groups is already conducted in Figure 1b.

4.       Figure 1: It is important to maintain consistency; the letters in the figure are lowercase, as are those referenced in the text. However, here they are written in uppercase, which is not uniform. Furthermore, Figure 1b is unclear and too small, making it difficult to discern the details in the figure caption.

5.       Line 153: I believe the title does not align with the descriptions in the figures, as they also include individuals without diabetes (ND), with diabetes but without obesity, etc. Please review and adjust the title to accurately reflect the content depicted.

6.       Line 159: Figure 2 is not referenced in the text.

7.       Figure 2: The quality and sharpness of the image are very poor; it needs improvement.

8.       Line 185-187: It is noteworthy that a title describes what is intended to be done. This title should be reviewed and rephrased; the specifics of the intended actions should be included in the results description. Also, what would be the difference from result 3.3? Isn't it the same? The distinction would lie in Table S3 of the supplementary data.

9.       Line 226: Be more specific and improve the wording regarding the reference to glucose and insulin, as it could be vague. Are you referring to serum concentrations, tolerance and resistance curves, or concentrations in the adipose tissue of the patients?

10.   Figure 3: The quality and sharpness of the figure need to be improved.

11.   Line 269: It would be highly important to present the data on glucose uptake in isolated adipocytes, ex vivo. Otherwise, it is challenging to perform a correlation analysis without understanding how individual uptake behaved.

12.   Line 272: Separate the words "The and lipids".

13.   Figure 6: Need to improve the quality and sharpness of the figure.

14.   Line 335: How does cholesterol increase lipid storage in the adipocyte?

Author Response

  1. Line 30: Due to the significance of the assertion and to ensure relevance to the current context, it would be advisable to update the citation, which is over 7 years old.

Response: Thank you for this suggestion. We have updated the citation to reflect more recent research on the global prevalence of Type 2 Diabetes (T2D) to https://www.who.int/news-room/fact-sheets/detail/diabetes (manuscript new reference 1).

  1. Table 1: Review the units in Table 1 for Fasting Plasma Glucose; they should be expressed as mmol/L.

Response: We apologize for the oversight. The units for Fasting Plasma Glucose in Table 1 have been corrected to mmol/L.

  1. Line 137: It would be clearer to specify that these analytes are from both individuals with T2D and ND, as the comparison between these groups is already conducted in Figure 1b.

Response: We agree with this suggestion. We have revised the text to clarify that the analytes are from both individuals with and without T2D (page 4 lines 142).

  1. Figure 1: It is important to maintain consistency; the letters in the figure are lowercase, as are those referenced in the text. However, here they are written in uppercase, which is not uniform. Furthermore, Figure 1b is unclear and too small, making it difficult to discern the details in the figure caption.

Response: Thank you for pointing out these issues. We have ensured consistency by using lowercase letters in both the figure and the text. Additionally, we have improved the clarity and size of Figure 1b.

  1. Line 153: I believe the title does not align with the descriptions in the figures, as they also include individuals without diabetes (ND), with diabetes but without obesity, etc. Please review and adjust the title to accurately reflect the content depicted.

Response: We have revised the title to accurately reflect the content depicted in the figures to “3.2. Metabolic Alterations in Adipose Tissue in Subjects with and without T2D and Obesity” (page 5, line 159).

  1. Line 159: Figure 2 is not referenced in the text.

Response: We apologize for the oversight. We have added a reference to Figure 2 in the text (page 5, lines 161-162).

  1. Figure 2: The quality and sharpness of the image are very poor; it needs improvement.

Response: We have improved the quality and sharpness of Figure 2 to ensure better readability.

  1. Line 185-187: It is noteworthy that a title describes what is intended to be done. This title should be reviewed and rephrased; the specifics of the intended actions should be included in the results description. Also, what would be the difference from result 3.3? Isn't it the same? The distinction would lie in Table S3 of the supplementary data.

Response: Thank you for this observation. We agree and have now merged this section into the section 3.3. (page 6)

  1. Line 226: Be more specific and improve the wording regarding the reference to glucose and insulin, as it could be vague. Are you referring to serum concentrations, tolerance and resistance curves, or concentrations in the adipose tissue of the patients?

Response: We have revised the text to specify that we are referring to concentrations of plasma glucose and serum insulin (page 8, lines 235-242).

  1. Figure 3: The quality and sharpness of the figure need to be improved.

Response: We appreciate your observation regarding Figure 3. We have ensured that the figures provided to the journal meet their quality specifications. It seems the PDF version reduced the quality. We will address this with the journal to ensure high-resolution figures are used in the final publication.

  1. Line 269: It would be highly important to present the data on glucose uptake in isolated adipocytes, ex vivo. Otherwise, it is challenging to perform a correlation analysis without understanding how individual uptake behaved.

Response: We acknowledge the importance of presenting glucose uptake data in isolated adipocytes. This data has been previously published (Pereira et al., Metabolism, 2016, reference 4 in manuscript). Nonetheless, the mean glucose uptake data is shown in Table 1 and referenced in the relevant publication for additional context (page 2, lines 76-77). Glucose uptake, both basal and insulin-stimulated is lower in subjects with T2D compared with subjects without.

  1. Line 272: Separate the words "The and lipids".

Response: We have corrected the typographical error (page 12, lines 282).

  1. Figure 6: Need to improve the quality and sharpness of the figure.

Response: See our response to comment 10.

  1. Line 335: How does cholesterol increase lipid storage in the adipocyte?

Response: Thank you for pointing out this issue, which has now been corrected (page 14, lines 352-354): “Elevated levels of specific lipids, such as certain fatty acids, have been shown to induce adipocyte hypertrophy and insulin resistance (reference 32 in manuscript).”

Reviewer 3 Report

Comments and Suggestions for Authors

This manuscript by Mathioudaki et al. studied the metabolomic profile of subcutaneous adipose tissues in people with diabetes and/or obesity. Although this study clearly suggests distinct metabolic dysregulation in T2D and obesity, there are several major concerns.

This study is overall descriptive and similar findings/conclusions have been reported in previous studies. There are many “assumptions” in the discussions, however, they are not purely based on the results of the current study.

The main figures are of low resolution and cannot present the results clearly. The adipocyte size and glucose uptake data were not presented to show the differences between study groups.

As previous studies have shown the metabolomic profiles in the plasma and visceral adipose tissues, it will be informative to compare these data with the current study to identify the tissue-specific metabolite signatures. 

Author Response

    1. This study is overall descriptive and similar findings/conclusions have been reported in previous studies. There are many “assumptions” in the discussions, however, they are not purely based on the results of the current study.

    Response: We acknowledge the descriptive nature of our study as a limitation, which is discussed on page 15, lines 422-423. To address this, we have revised the discussion section to minimize assumptions and clearly differentiate between findings directly supported by our data and those that are interpretations. However, studies investigating untargeted metabolomics in subcutaneous adipose tissue from subjects with and without T2D and obesity are lacking. Therefore, we consider that even if our study is descriptive, it adds new knowledge to the field. These considerations are included in the discussion (page 13, lines 313-315).

    1. The main figures are of low resolution and cannot present the results clearly.

    Response: Thank you for this comment which was also noted by reviewer 2. We have ensured that the figures provided to the journal meet their quality specifications. It seems the PDF version reduced the quality. We will address this with the journal to ensure high-resolution figures are used in the final publication.

    1. The adipocyte size and glucose uptake data were not presented to show the differences between study groups.

    Response: The lack of glucose uptake was also noted by reviewer 2. We acknowledge the importance of presenting glucose uptake data in isolated adipocytes. This data has been previously published (Pereira et al., Metabolism, 2016, reference 4 in manuscript). Nonetheless, the mean glucose uptake data is shown in Table 1 and referenced in the relevant publication for additional context (page 2, lines 76-77). Glucose uptake, both basal and insulin-stimulated is lower in subjects with T2D compared with subjects without. see the response to reviewer 2, comment 11. Cell size data is shown in Table 1 and no significant differences were found between BMI-matched subjects with and without T2D.

    1. As previous studies have shown the metabolomic profiles in the plasma and visceral adipose tissues, it will be informative to compare these data with the current study to identify the tissue-specific metabolite signatures. 

    Response: Subjects with both T2D and obesity exhibited significant metabolic alterations in adipose tissue, highlighted by elevated levels of amino acids (including BCAAs like leucine, and isoleucine), peptides, carbohydrates (e.g., glucose, mannose), and energy-related metabolites (e.g., alpha-ketoglutarate, malate, gluconate). These findings are in line with earlier studies in plasma that have identified increased levels of BCAAs, aromatic amino acids, and disrupted lipid metabolism, alongside elevated glycolytic intermediates and TCA cycle metabolites [1-3]. The association of elevated BCAAs with diabetes and insulin resistance in plasma and skeletal muscle has been well-documented for decades [4-8], typically linked to impaired catabolism. Previous studies in VAT and SAT have also shown elevated levels of amino acids including BCAAs, tryptophan and serine in subjects with hyperglycemia and T2D [8-10].

    Interestingly, some metabolic signatures found in SAT in this study are novel, such as elevated levels of the TCA intermediates (alpha-ketoglutarate, malate, and gluconate) and reduced nucleotide metabolites (adenine, adenosine, AMP, and CMP), which have not been reported in previous adipose tissue metabolomics studies in T2D subjects. Changes in these metabolites can impact energy metabolism [11, 12], and contribute to metabolic dysregulation observed in T2D.

    We have added this comparison to the discussion section (page 13, lines 324-340).

    References

    [1] Kučera J, Spáčil Z, Friedecký D, Novák J, Pekař M, Bienertová-Vašků J. Human White Adipose Tissue Metabolome: Current Perspective. Obesity (Silver Spring). 2018;26:1870-8.

    [2] Newgard CB. Interplay between lipids and branched-chain amino acids in development of insulin resistance. Cell Metab. 2012;15:606-14.

    [3] Shahisavandi M, Wang K, Ghanbari M, Ahmadizar F. Exploring Metabolomic Patterns in Type 2 Diabetes Mellitus and Response to Glucose-Lowering Medications—Review. Genes. 2023;14:1464.

    [4] Cuomo P, Capparelli R, Iannelli A, Iannelli D. Role of Branched-Chain Amino Acid Metabolism in Type 2 Diabetes, Obesity, Cardiovascular Disease and Non-Alcoholic Fatty Liver Disease. Int J Mol Sci. 2022;23.

    [5] Vanweert F, Schrauwen P, Phielix E. Role of branched-chain amino acid metabolism in the pathogenesis of obesity and type 2 diabetes-related metabolic disturbances BCAA metabolism in type 2 diabetes. Nutr Diabetes. 2022;12:35.

    [6] Ahola-Olli AV, Mustelin L, Kalimeri M, Kettunen J, Jokelainen J, Auvinen J, et al. Circulating metabolites and the risk of type 2 diabetes: a prospective study of 11,896 young adults from four Finnish cohorts. Diabetologia. 2019;62:2298-309.

    [7] Wildberg C, Masuch A, Budde K, Kastenmüller G, Artati A, Rathmann W, et al. Plasma Metabolomics to Identify and Stratify Patients With Impaired Glucose Tolerance. J Clin Endocrinol Metab. 2019;104:6357-70.

    [8] Diamanti K, Cavalli M, Pan G, Pereira MJ, Kumar C, Skrtic S, et al. Intra- and inter-individual metabolic profiling highlights carnitine and lysophosphatidylcholine pathways as key molecular defects in type 2 diabetes. Sci Rep. 2019;9:9653.

    [9] Diamanti K, Visvanathar R, Pereira MJ, Cavalli M, Pan G, Kumar C, et al. Integration of whole-body [(18)F]FDG PET/MRI with non-targeted metabolomics can provide new insights on tissue-specific insulin resistance in type 2 diabetes. Sci Rep. 2020;10:8343.

    [10] Piro MC, Tesauro M, Lena AM, Gentileschi P, Sica G, Rodia G, et al. Free-amino acid metabolic profiling of visceral adipose tissue from obese subjects. Amino Acids. 2020;52:1125-37.

    [11] Sharma NK, Das SK, Mondal AK, Hackney OG, Chu WS, Kern PA, et al. Endoplasmic reticulum stress markers are associated with obesity in nondiabetic subjects. J Clin Endocrinol Metab. 2008;93:4532-41.

    [12] Liu K, Jin X, Zhang X, Lian H, Ye J. The mechanisms of nucleotide actions in insulin resistance. Journal of Genetics and Genomics. 2022;49:299-307.

Round 2

Reviewer 3 Report

Comments and Suggestions for Authors

The manuscript has been revised appropriately per prior comments. However, the figures remain low-resolution.